

# Strategies of elite Chinese gymnasts in coping with landing impact from backward somersault

Chengliang Wu[1,2], Weiya Hao[3], Qichang Mei[4], Xiaofei Xiao[5], Xuhong Li[6] and Wei Sun[7]

[1] School of Kinesiology, Shanghai University of Sport, Shanghai, China
[2] School of Physical Education and Health, Chongqing Three Gorges University, Chongqing, China
[3] China Institute of Sport Science, Beijing, China
[4] Auckland Bioengineering Institute, University of Auckland, Auckland, New Zealand
[5] School of Rehabilitation Medicine, Binzhou Medical University, Yantai, China
[6] School of Physical Education and Health, Hangzhou Normal University, Hangzhou, China
[7] Sports Biomechanics Lab, Shandong Institute of Sports Science, Jinan, China

Corresponding author
Weiya Hao, haoweiya@ciss.cn

## ABSTRACT

This study aimed to investigate how elite Chinese gymnasts manage the landing impact from a backward somersault. Six international-level male gymnasts performed backward somersault tests with a synchronous collection of kinematics (250 Hz), ground reaction forces (1,000 Hz), and surface electromyography (EMG) (2,000 Hz). A 19-segment human model was developed and lower extremity joints torques were calculated by means of a computer simulation. The angles of the lower extremity joints initially extended and then flexed. These angular velocities of extension continued to decrease and the joint torques changed from extensor to flexor within 100 ms before touchdown. The angles of the hips, knees, and ankles flexed rapidly by 12°, 36°, and 29°, respectively, and the angular velocities of flexion, flexor torque, and EMG peaked sharply during the initial impact phase of the landing. The angles of the hips, knees, and ankles flexed at approximately 90°, 100°, and 80°, respectively. The torques were reversed with the extensor torques, showing a relatively high level of muscle activation during the terminal impact phase of the landing. The results showed that the international-level gymnasts first extended their lower extremity joints, then flexed just before touchdown. They continued flexing actively and rapidly in the initial impact phase and then extended to resist the landing impact and maintain body posture during the terminal impact phase of the landing. The information gained from this study could improve our understanding of the landings of elite gymnasts and assist in injury prevention.

## INTRODUCTION

Gymnastics is a popular sport with 50 million participants worldwide (*Slater et al., 2015*). Each gymnastics routine ends with a landing, and successful landings (without taking a step or falling) are a key factor for motion evaluation. However, there is a high incidence of
injury reported in gymnasts while landing, especially to the lower extremity joints (*Westermann et al., 2015*). A better understanding of the potential injury mechanisms could help prevent certain injuries and change the design of the training schemes, thereby improving performance.

Gymnasts were reported to bear high frequency (over 200 times a week) landing impact loads (*Gittoes & Irwin, 2012*), with the peak vertical ground reaction force (vGRF) reaching 7.1–15.8 times the athlete's body weight (BW) (*Slater et al., 2015*). This repetition and large GRF has caused a high rate of lower extremity injuries in gymnasts (*Mills, Pain & Yeadon, 2009*). It has also been suggested that there is a correlation between the high injury incidence and the excessive load to the lower extremities of gymnasts (*Daly, Bass & Finch, 2001*; *Wade et al., 2012*). Furthermore, the Code of the International Gymnastics Federation (FIG) requires that there be no excessive knee flexion during landing from gymnasts for aesthetic purposes (*FIG, 2017*). Several studies demonstrated that gymnasts produced greater peak vGRF than recreational athletes in drop landings, which are considered the stiff lower extremity landing techniques, with knee flexion less than 90° (*Christoforidou et al., 2017*; *Devita & Skelly, 1992*; *Seegmiller & McCaw, 2003*). More specifically, this landing pattern increased leg stiffness and is a potential contributing factor for injury (*Butler, Crowell & Davis, 2003*). *Bradshaw & Hume (2012)* investigated the changes in the landings of two gymnasts over a period of 8 years. Both gymnasts showed an increased ankle plantar-flexion stiffness by 10.8 and 13.9 kN/m, respectively, with both gymnasts reporting severe pain in one or both heels over this period of time. Furthermore, other contributing factors to injuries in these gymnasts, including internal factors such as anatomical differences, neuromuscular function, strength, and leg stiffness, were observed. External factors, such as the landing of complex tasks, exposure time, the training environment, and varied competition were considered (*Bradshaw & Hume, 2012*).

Few studies have attempted to explore strategies for reducing the impact force and the incidence of injury. One study found that increasing the flexion of the lower extremity joints during landing could effectively reduce the impact load (*Slater et al., 2015*). However, this may lead to compensatory muscle and ligament injuries, thus affecting the landing stability, especially during a high-speed landing impact (*Bradshaw & Hume, 2012*; *Tant, Wilkerson & Browder, 1989*). Modifying the material composition of the gymnastics landing mat may reduce the vGRF, but increase the internal load (muscle forces and joint reaction forces) to the lower extremity joints (*Mills, Pain & Yeadon, 2009*) and the potential for subtalar and ankle instability during the landing (*McNitt-Gray, 2000*). It is worth noting that previous studies generally analyzed the whole impact phase of the landing, defined from the initial ground contact to the maximal knee flexion (*Christoforidou et al., 2017*), the maximal descending height of the center of mass (*Caine et al., 2003*), or the local minima in the vGRF (*McNitt-Gray et al., 2001*). Participants in these studies were collegiate students or young athletes who did not participate in international- or national-level gymnastics competitions. Therefore, it is unclear what strategies elite international-level gymnasts used and whether they adapted initial (from the touchdown to the peak vGRF) and terminal impact-phase (from the peak vGRF return to their BW) strategies for the flexion/extension of the lower extremity

joints before the touchdown of the landing. Moreover, the lower extremity joints approach full extension and then flexion just before touchdown during the flight phase of drop landings (*McNitt-Gray, Yokoi & Millward, 1993*). However, it is not clear whether the same strategies would be seen before the touchdown of landings from other completed tasks in gymnastics. Using an in-vivo implanted sensor to test the internal load of the human lower extremity would be useful but there are ethical limitations to this methodology. A computer simulation of the human body, however, provides a practical approach to explore the characteristics of body motion and has been widely used to analyze body movement in humans (*Pandy, 2001*).

The backward somersault (BS) is one of the most basic and common movements for gymnasts for developing difficulty movement and combined motion, and is used frequently in gymnastics training and competitions. The aim of this study was to investigate how elite gymnasts manage the landing impact from a BS and achieve a safe, aesthetic, and stable performance. It is our hypothesis that elite gymnasts utilize varied flexion/extension strategies to control their lower extremity joints during the different landing phases (before touchdown, initial, and terminal impact-phases of the landing) of the BS. Therefore, a study of the basic movements would help us understand the characteristics and injury mechanisms of the more complex gymnastics landing of the same type.

## MATERIALS AND METHODS

### Participants

Six international-level male gymnasts from the Chinese national team competing in World Cups and/or Championships, with no musculoskeletal injuries for at least 6 months prior, participated in the study (mean ± standard deviation (SD) age: 17.3 ± 1.3 years, height: 165.7 ± 5.0 cm, body mass: 57.3 ± 3.9 kg). All participants were familiarized with the procedures in advance and informed consents were signed. The study was approved by the Ethical Advisory Committee of the China Institute of Sport Science (WEI 16-27) in accordance with the regulations set forth by the Declaration of Helsinki.

### Procedure

The experiment was conducted in the biomechanics laboratory of the China Institute of Sport Science. A 9-camera Qualisys Oqus motion system (250 Hz, Gothenburg, Sweden) was used to capture the 3D motion data. The standard reflective markers (diameter: 16 mm) were placed at the head, cervical vertebrae (CV7), scapula-inferior angle, thoracic vertebrae (TV10), shoulder, elbow, wrist, anterior superior iliac spine, posterior superior iliac spine, knee, ankle, metatarsal-phalangeal joints, heel, and toes on both sides of the body (Figs. 1A–1C). The marker placements were referenced from the CAST full body marker set (*Sint, 2007*). A Kistler force plate (400 × 600 mm), located beneath a landing mat (five cm thick), was used to collect vGRF data (1,000 Hz) and was surrounded by an ethylene-vinyl acetate insole mat. The vGRF was reported to decrease by 5% on the force plate when the landing mats were up to 12 cm thick (*McNitt-Gray et al., 2001*). Surface muscle activity signals were recorded (2,000 Hz, Bagnoli 8 Desktop

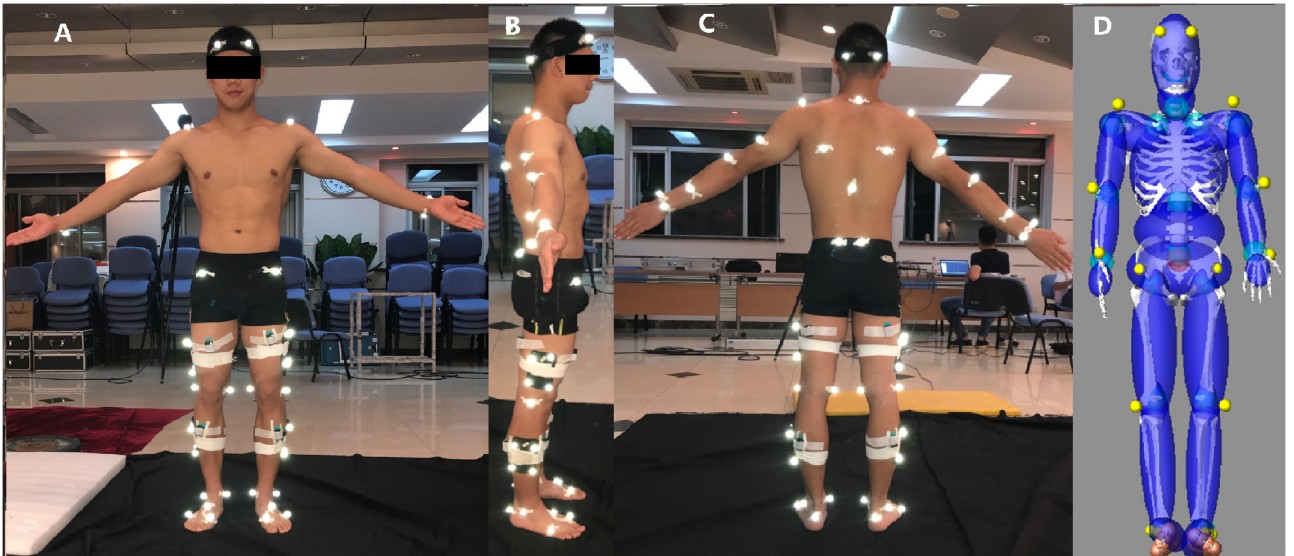

**Figure 1 Location of retro-reflective markers and EMG sensors on the gymnasts (A–C), 19 segments human model generated by GEBOD in LifeMOD™ (D).**

electromyography (EMG) System; Delsys Inc, Boston, MA, USA) from the rectus femoris (RF), biceps femoris (BF), tibialis anterior (TA), and lateral gastrocnemius (LG) on the two lower extremities of the gymnasts as per the SENIAM guidelines (*Hermens et al., 1999*). The Qualisys Oqus motion capture system, the Kistler force plate, and the surface EMG system were all synchronized during the data collection.

The gymnasts completed an initial warm-up exercise (15 min of jogging, jumping, and stretching) and then each participant performed three successful trials of BS without taking a step or hop. The BS was initiated by jumping from the ground next to the force plate with bare feet (Fig. 2). The best trial for each participant was chosen by two national-level judges using the Code of Points of FIG for further analysis.

## Experiment data reduction and analysis

The 3D motion data was processed using Qualisys Track Manager Software, following a 10 Hz low-pass cut-off filter (*Slater et al., 2015*). The joint angles with its angular velocity were calculated between two lines in space based on three-dimension trajectories. The vGRFs were filtered using a low-pass cut-off at 50 Hz (*Slater et al., 2015*). The peak vGRFs were normalized with the BW of each gymnast. The touchdown was identified as the first frame when vGRF exceeded 10 N (*Christoforidou et al., 2017*). Raw EMG signals were fully wave rectified and band-pass filtered by 10–400Hz (*Van Dieën et al., 2009*).

The pre-activation phase (T0) was defined as the 100 ms duration before touchdown (*Komi & Bosco, 1987*). The EMG processing was conducted in the pre-activated and two impact phases of the landing (T1: initial impact-phase, from contact to the peak vGRF; T2: terminal impact-phase, from the peak vGRF return to their BW) (Fig. 2). The EMG signals were normalized by the peak EMG of each trial. Antagonist–agonist co-activation was calculated as TA to LG normalized EMG for ankle, and BF to RF normalized

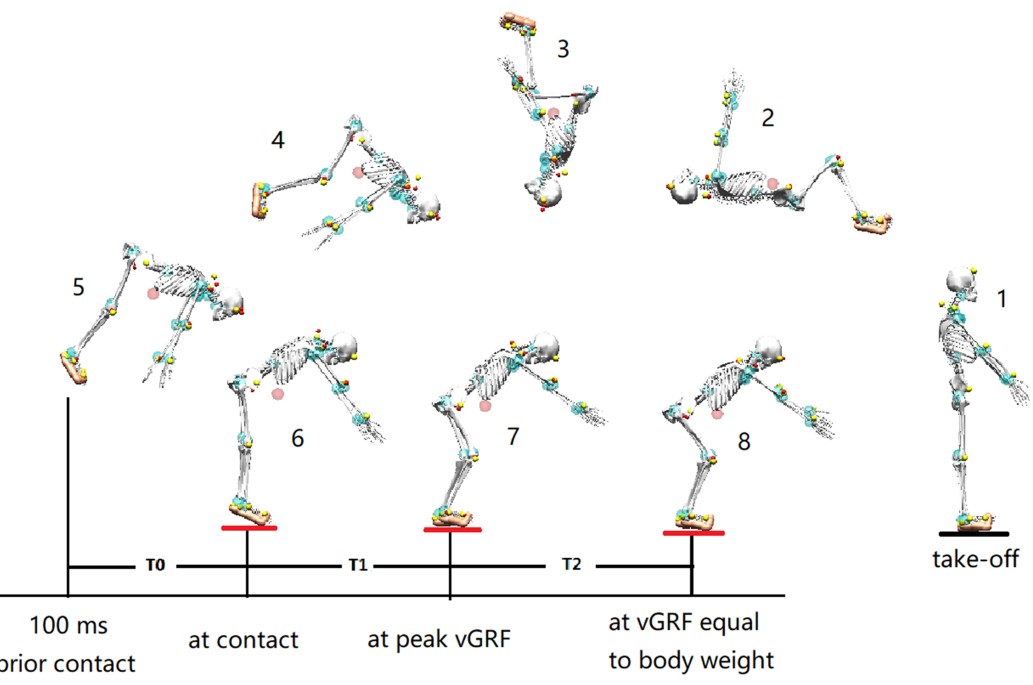

**Figure 2 The demonstration of backward somersault landing.** T0: The pre-activation phase was defined as 100 ms preceding ground contact; T1: initial impact-phase, from the first touchdown to the peak vertical ground reaction force (vGRF); T2: terminal impact-phase, from the peak vGRF to the vGRF equaling to body weight.

EMG for knee (*Ruan & Li, 2010*). All of the experimental data for the gymnasts were averaged within 20 ms intervals and then the data was reported as mean ± SD using descriptive statistics.

## Computer simulation and validation

The experimental findings from the elite gymnasts showed a low discrete degree, with the mean standard error within six participants of 0.6 BW (peak vGRF), 2.7 ms (time to peak vGRF) and ranging from 2° to 10° in the lower extremity angles. One gymnast with intermediate experiment results was chosen for modeling and simulation.

LifeMod (LifeModeler, Inc., San Clemente, CA, USA) is an advanced multi-body computer simulation software system commonly used in human movement simulation with Automatic Dynamic Analysis of Mechanical Systems (ADAMS) as the dynamics modeling engine. The GeBod database (BRG.LifeMOD$^{TM}$) was used to develop a 19-segment and 50 degree of freedom rigid-body model based on age (17 years), weight (63 kg), and height (1.68 m) data from the selected gymnast. The model consisted of the head, neck, upper torso, central torso, lower torso, scapulas, upper arms, lower arms, hands, upper legs, lower legs, and feet (Fig. 1D) (*Serveto et al., 2010*). A model of the gymnastics matting with a dimension of 2 × 2 × 0.05 m (length × width × height) was developed using MSC.ADAMS (MSC Software Corp. acronym of Automated Dynamic Analysis of Mechanical Systems) software. The basis for the mechanical properties of the landing mat was obtained by an optimization algorithm. The model was then validated by

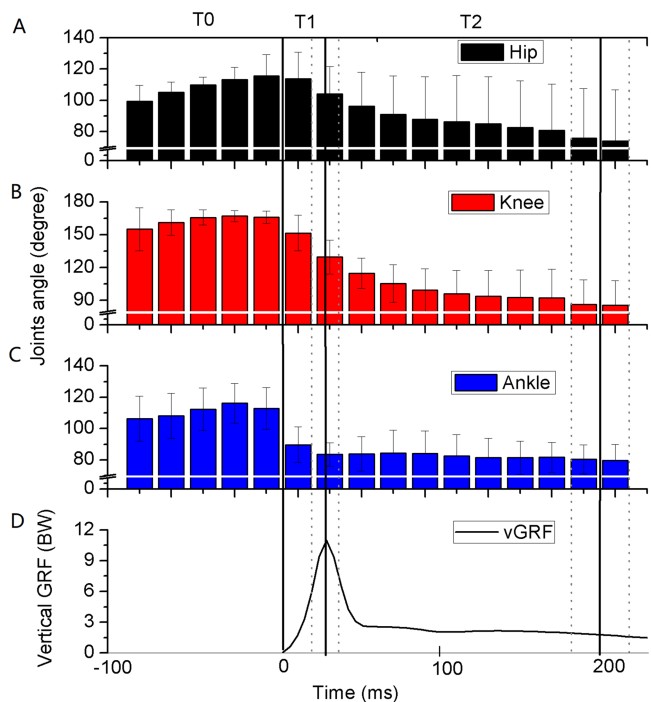

**Figure 3** **Average integrated (20 ms bins) angles of lower extremity joints (A–C) and vertical ground reaction force (vGRF) (D) during backward somersault landing (_n_ = 6).** The whole landing process was divided into three phases by solid vertical lines. T0: The pre-activation phase was defined as 100 ms preceding ground contact; T1: initial impact-phase, from the first touchdown to the peak vGRF; T2: terminal impact-phase, from the peak vGRF to the vGRF equaling to body weight (Mean: solid vertical line, standard deviation: dotted vertical line).

coefficients of multiple correlations (CMC) between the simulation and actual results, with the specific algorithm described in detail in our previous study (_Xiao et al., 2017_). After verifying the reliability of the model, the joint torques of the hip, knee, and ankle joints were conducted using computer simulation.

## RESULTS

The measured angles of the hip, knee, and ankle joints were first extended (mean 11°, 10°, and 6°, respectively) and then flexed during T0 in the six gymnasts (Figs. 3A–3C). The corresponding joint angles rapidly flexed by 2°, 36°, and 29° during T1 and maintained at around 90°, 100°, and 80° during T2, respectively. The angular velocities of extension continued to decrease during T0, and the angular velocities of flexion reached their peaks during T1 and gradually approached zero during T2 (Figs. 4A–4C). The eight muscles in the bilateral lower extremity were pre-activated during T0 (Figs. 5A–5D). The EMG amplitude of most muscles increased from T0 to T1, and reached their maximum near the peak vGRF. They still maintained a high-level of activation during T2.

The CMC between the measured lower extremity joint angles and the simulation result were calculated for model validation, with the left knee (CMC = 0.95), right knee (CMC = 0.93), and left and right ankles (CMC = 0.85) (Figs. 6A and 6B). The CMCs

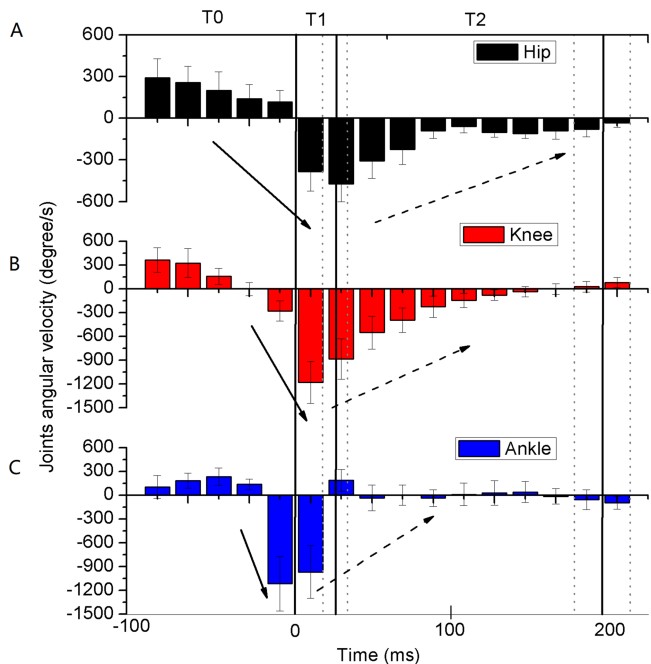

**Figure 4 Average integrated (20 ms bins) angular velocities of lower extremity joints during backward somersault landing (A–C) ($n = 6$).** The solid arrow indicates that the acceleration is negatively increasing, and the dotted arrow indicates that the acceleration is negatively decreasing. The whole landing process was divided into three phases by solid vertical lines. T0: The pre-activation phase was defined as 100 ms preceding ground contact; T1: initial impact-phase, from the first touchdown to the peak vertical ground reaction force (vGRF); T2: terminal impact-phase, from the peak vGRF to the vGRF equaling to body weight (Mean: solid vertical line, standard deviation: dotted vertical line).

greater than 0.75 indicated good correlations (*Collins et al., 2009*). The difference between the simulated (11.9 BW) and measured peak vGRF (12.5 BW) was 4.6%, which was less than 10% and was considered to be an accurate representation (*King, Wilson & Yeadon, 2006*). Therefore, these results could be used to validate the model. The angles and angular velocities of the lower extremity joints were consistent with the results of Figs. 3A–3C and 4A–4C, respectively (Figs. 6A–6D). The torques of the lower extremity joints were initially dominated by extensor during T0 (plantar flexor for the ankle) (Figs. 6E and 6F), and reversed to flexor, reaching their maximum (dorsiflexion for the ankle) during T1. The torques quickly reversed again to the extensor and reached their peak during T2 (plantar flexor for the ankle). The torque of the knee joints reached their maximum and in the ankle joints the torques remained slight.

## DISCUSSION

From the best of our knowledge, this is the first investigation to reveal the flexion/extension strategies of the lower extremity joints from flight to the initial and terminal impact-phases of BS landings in elite gymnasts. The study quantified the kinematics, kinetics, and muscle activation characteristics of each phase of the BS in elite gymnasts. The findings could enhance our understanding of the gymnastics landing.

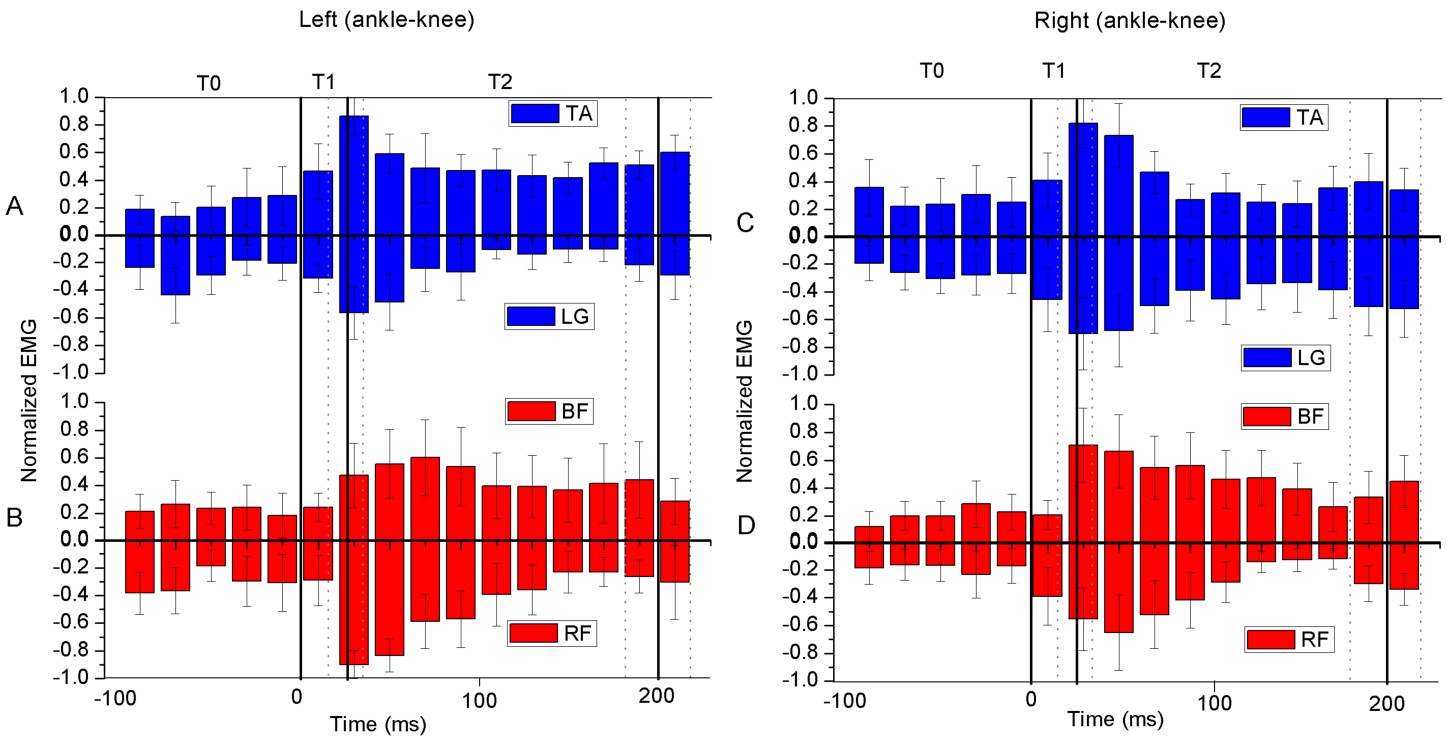

**Figure 5 Average integrated (20 ms bins) coactivation of root mean square of normalized EMG (EMG$_{RMS}$) ($n = 6$).** Positive EMG$_{RMS}$ means antagonist of the joint, and negative EMG$_{RMS}$ means the agonist of the joint. The EMGs were normalized to the peak EMG during the landing phase. RF, rectus femoris; BF, biceps femoris; TA, tibialis anterior; LG, lateral gastrocnemius. The coactivation was defined as the TA to LG EMG for the ankle (A and C), and the BF to RF EMG for the knee (B and D). The whole landing process was divided into three phases by solid vertical lines. T0: The pre-activation phase was defined as 100 ms preceding ground contact; T1: initial impact-phase, from the first touchdown to the peak vertical ground reaction force (vGRF); T2: terminal impact-phase, from the peak vGRF to the vGRF equaling to body weight (Mean: solid vertical line, standard deviation: dotted vertical line).

The lower extremity joints of the gymnasts first extended and then flexed actively during the preparation phase for the touchdown (T0). Increased angles of the lower extremity joints were conducive to the body's extension, which may contribute to increasing the moment of inertia of the body. As the moment of inertia increases, the gymnasts can reduce their angular velocity in preparation for touchdown (*McNitt-Gray, 2000*). It was noteworthy that the angular velocities of the knee and ankle joints changed from extension to flexion before touchdown. The lower extremity joints actively flexed, which was first seen in the ankle joint, followed by the knee and hip joints. *Gittoes et al. (2011)* suggested that the female gymnasts' knees and ankles were also flexing at the touchdown of the backward rotating pike and tuck dismounts, but there was no information about whether the joints flexed before touchdown. Furthermore, the antagonist and agonist muscles of the lower extremity joints were pre-activated, which may play an important role in regulating the lower extremity stiffness (*Christoforidou et al., 2017*). Muscle contraction is an essential factor for producing lower extremity joint torque in the flight before landing. The lower extremity joints experienced the process from extension to flexion because these torques changed from extensor to flexor. Previous studies have focused on the torque of the lower extremity joints after touchdown (*McNitt-Gray et al., 2001*;

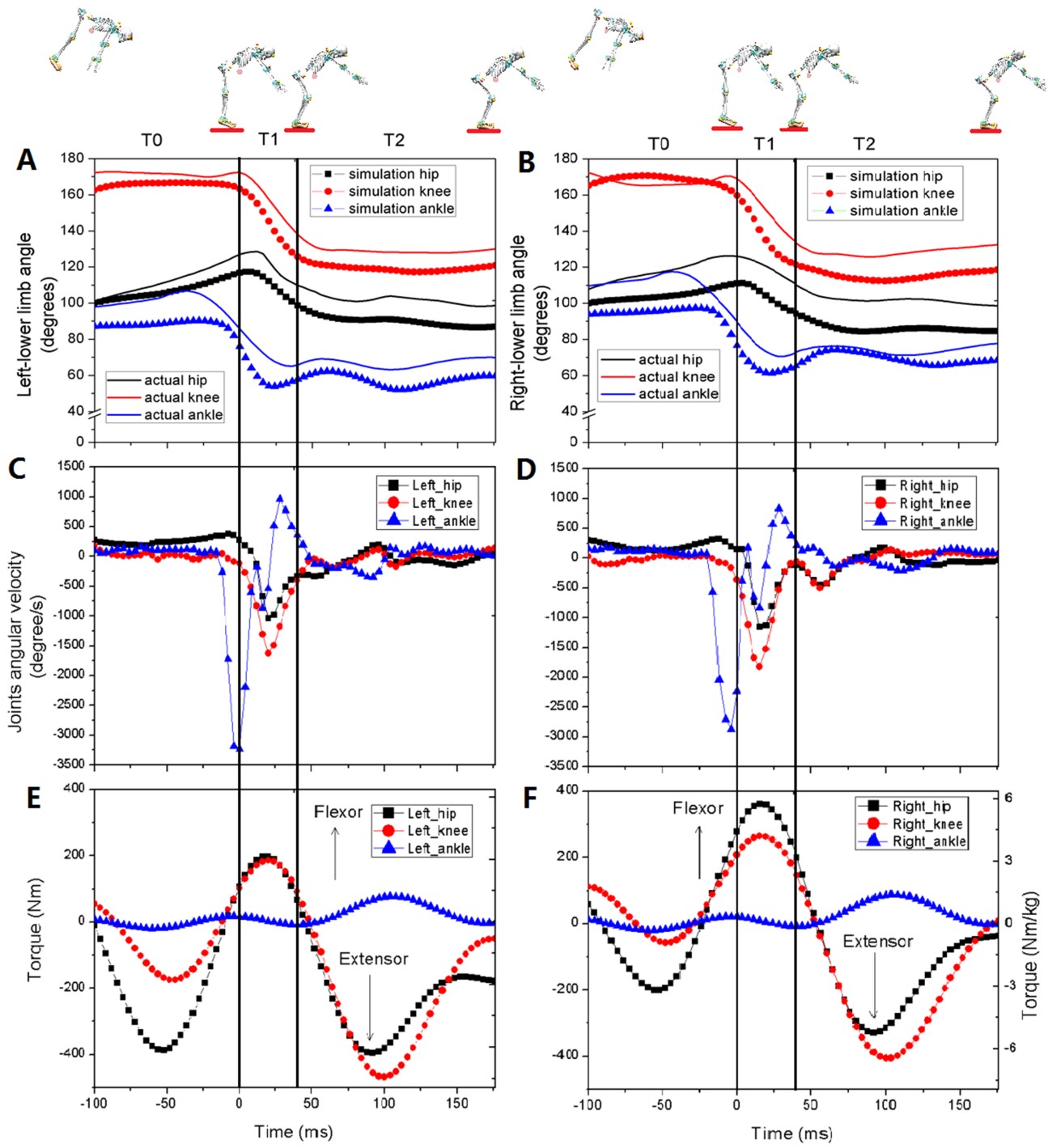

**Figure 6 The angles (A and B), angular velocities (C and D) and torques (E and F) of lower extremity joints for one of the gymnasts during backward somersault landing ($n = 1$).** The skeleton models showed four body postures of the landing (100 ms prior touchdown, touchdown, peak vertical ground reaction force (vGRF), and vGRF equal to body weight, respectively). The whole landing process was divided into three phases by solid vertical lines. T0: The pre-activation phase was defined as 100 ms preceding ground contact; T1: initial impact-phase, from the first touchdown to the peak vGRF; T2: terminal impact-phase, from the peak vGRF to the vGRF equaling to body weight.

*Mills, Pain & Yeadon, 2009*; *Verniba et al., 2017*), but there is little knowledge about the torque of the joints in the pre-landing phase. Our results indicated that in preparation for the landing, the lower extremity joints of the gymnast would first be extended and then actively flexed just before touchdown.

During the initial impact-phase of the landing (T1), the lower extremity joints continue flexing actively and rapidly. T1 has a very short duration ($22.8 \pm 6.7$ ms), which was consistent with the results of a previous study (*Slater et al., 2015*) where the angles of the lower extremity joints flexed rapidly. Furthermore, the angular velocities of the hip joints changed from extension to flexion and the angular velocities of the knee and ankle joints further flexed and reached their peak values. It is significant to note that the peak angular velocities of the ankle joint occurred before touchdown, but the peak angular velocities of the knee joint occurred before the peak of the vGRF and the peak angular velocities of the hip joint occurred near the peak vGRF. They reached peak values successively, which may be because the ankle is the initial interface with the ground, and the hip is the proximal joint, which properly reflects the synergy of multiple joints (*Gittoes et al., 2011*). *McNitt-Gray et al. (2001)* suggested that only the flexor torques of the hip joints were generated in a short time (about 20 ms) after touchdown and the other joints of the lower extremities maintain extensor torques during the landing. However, our results confirmed that all torques of the lower extremity joints were flexor torques in T1. This difference in lower extremity joint torques might be due to the landing being performed by different levels of participants (collegiate male gymnasts in their study against the international-level gymnast in this study). Therefore, the authors speculated that the participants (international-level gymnasts) in this study would represent an altered landing control strategy to help improve performance. Additionally, the results of muscle activation showed that the EMG amplitudes of the antagonistic and agonistic muscles of the lower extremity joints were close to reaching their peak values synchronously, and therefore this regulated the lower extremity stiffness to accommodate the rapidly increased vGRF (*Kramer et al., 2012*). A previous study indicated that the activity levels of the lower extremity muscles positively correlated with the peak vGRF during the landing absorption phase, which contributed to absorbing the impact of the landing and preventing injury to the lower extremities (*Iida et al., 2011*). Therefore, in order to quickly absorb the landing impact and prevent injuries during T1, the authors suggest that the lower extremity joints of gymnasts may flex continuously and actively, while increasing the activity levels of the lower extremity muscles.

During the terminal impact-phase of the landing (T2), the gymnasts in this study began to actively extend their lower extremity joints to resist the impact force and maintain body posture. The angles and angular velocities of the lower extremity joints flexed minutely. Finally, the lower extremity joints stabilized and maintained a small flexion, thus these gymnasts met the Code of Points of FIG, which requires that the joints should not be excessively flexed (*FIG, 2017*). Furthermore, the EMG of the lower extremity muscles remained at a high level of activation, and the extensor torques of the lower

extremity joints (plantar flexion torque of ankle) increased simultaneously, reaching their peak values during T2. Therefore, accomplishing the landing task with a small flexion of the lower extremity joints may allow the gymnasts to generate the joint torques needed to compensate for the large vGRF. Appropriate muscle length may be the key to producing greater joint torques (*McNitt-Gray, 2000*). *Marchetti et al. (2016)* found that the highest overall muscle activation of the lower extremities was generated at a 90° knee joint angle during a back squat, which was close to the knee angles of this study during T2.

*Devita & Skelly (1992)* suggested that a stiff landing in which there is less than 90° of knee flexion had larger GRFs than a soft landing; the ankle plantar flexors produced a larger torque, and the ankle muscles absorbed a greater amount of the impact forces upon landing. The ankle is more likely to be injured in this condition, and this was consistent with the findings of an epidemiological investigation (*Kerr et al., 2015*). The study indicated that the gymnasts first extend their lower extremity joints before touchdown, then flex actively and rapidly in anticipation of the upcoming touchdown and the initial impact phase, and again extend during the terminal impact phase of the landing. These landing strategies could effectively mitigate some of the landing impact and are conducive to injury prevention.

There are a few limitations in this study. First, this study is limited to a population of international-level male gymnasts, but we acknowledge that there might be differences in lower extremity kinematics and kinetics between different genders during their landings (*Haines et al., 2011*). Secondly, the sample size (six international-level gymnasts) of this study is limited and the authors did not consider more gymnasts of different levels, which was the original intent in order to control variables and reveal the landing strategies primarily in international-level gymnasts. However future studies should include male and female gymnasts of different levels.

## CONCLUSIONS

The study quantified the lower extremity kinematics, kinetics, and muscle activation of international-level gymnasts during BS landings. Gymnasts first extend their lower extremity joints to increase the moment of inertia, thus better reducing the body's angular velocity before touchdown. The lower extremity joints flex actively just before touchdown, continue flexing actively and rapidly in the initial impact phase, and then extend to resist the impact force and maintain body posture during the terminal impact phase of the landing. This is the first study to investigate altered flexion/extension strategies of the lower extremity joints during different phases of the landing in gymnastics, thereby having the potential to expand the current understanding of the landing process of gymnastics and to aid in the prevention of injuries.

## ACKNOWLEDGEMENTS

The authors are thankful to the Chinese national gymnastics team for their valuable collaboration.

### Funding

This work was supported by the National Natural Science Foundation of China (Nos. 11672080, 31700815), the National Key R&D Program of China (No. 2017YFC0803802), the Chongqing Natural Science Foundation of China (No. cstc2018jcyjAX0086), and Shandong Natural Science Foundation of China under Grant (No. ZR2018LA011). The funders had no role in study design, data collection and analysis, decision to publish, or preparation of the manuscript.

### Grant Disclosures

The following grant information was disclosed by the authors:
National Natural Science Foundation of China: 11672080, 31700815.
National Key R&D Program of China: 2017YFC0803802.
Chongqing Natural Science Foundation of China: cstc2018jcyjAX0086.
Shandong Natural Science Foundation of China under Grant: ZR2018LA011.

### Competing Interests

The authors declare that they have no competing interests.

### Author Contributions

- Chengliang Wu conceived and designed the experiments, performed the experiments, analyzed the data, contributed reagents/materials/analysis tools, prepared figures and/or tables, authored or reviewed drafts of the paper, approved the final draft.
- Weiya Hao analyzed the data, prepared figures and/or tables, authored or reviewed drafts of the paper, approved the final draft.
- Qichang Mei analyzed the data, authored or reviewed drafts of the paper, approved the final draft.
- Xiaofei Xiao performed the experiments, contributed reagents/materials/analysis tools, prepared figures and/or tables, authored or reviewed drafts of the paper, approved the final draft.
- Xuhong Li performed the experiments, contributed reagents/materials/analysis tools, prepared figures and/or tables, authored or reviewed drafts of the paper, approved the final draft.
- Wei Sun analyzed the data, authored or reviewed drafts of the paper, approved the final draft.

### Human Ethics

The following information was supplied relating to ethical approvals (i.e., approving body and any reference numbers):

The Ethical Advisory Committee of the China Institute of Sport Science granted Ethical approval to carry out the study within its facilities (Ethical Application Ref: WEI 16-27).

## Data Availability

The raw measurements are available in the Supplemental Files.

## Supplemental Information

Supplemental information for this article can be found online at http://dx.doi.org/10.7717/peerj.7914#supplemental-information.

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
