# Peer review of "Strategies of elite Chinese gymnasts in coping with landing impact from backward somersault"

_PeerJ, doi:10.7717/peerj.7914_

## Round 0.1 · original submission · Major Revisions

There have been some diverging comments from the three reviewers on this manuscript, and therefore I feel it's most appropriate that the authors are given the opportunity to revise the manuscript with respect to these constructive criticisms of the reviewers. In particular, please pay particular attention to the comments of reviewer one and reviewer two who both currently have major reservations with the manuscript.


[]

·

Basic reporting

This is a descriptive study about landing after a backward somersault assessed in 6 international level gymnasts. It is an interesting subject and challenging to assess.

Although I'm also not a native English speaker, the text has serious issues regarding the language. It was really hard to read and in many cases hard to understand what is meant. Since my comprehension of this manuscript is very low in its current form, it does not allow me to fully judge its contents, especially regarding its scientific aspect.

Apart from the above authors the authors make a new review of the literature in the introduction, but this does not converge to the subject of the study. There are serious concerns regarding the rational of the study, and there is no clear justification of the scientific approach followed by the authors. Although I understand that this is mainly a descriptive study, the authors should try to build some basic hypothesis, based on the current findings.

Experimental design

The research question is not clearly described and not connected with the conclusions.
For example, the last sentence of the introduction states that "knowledge from this study would be used for the identification and selection of potential young gymnast athletes". How? The last sentence of the conclusions reads "Knowledge from this study would be used for the identification and selection of potential young gymnast athletes". How?

Reading the methods it is very hard to reproduce this experiment, due to language issues but not only (especially regarding the simulation).
It is not clear what was the scope of the simulation. There is no clear connection of this approach in the introduction and conclusions.
There is no description of the statistics.

Validity of the findings

There are many concerns - that I cannot cite all of them - regarding the validity of the findings.
It is not clear in the figures whether the presented data are from a single subject or from all subjects.
Even in Figure 6 it is not clear whether this is one trial or the average from more. What does the left and right column of figures represent?
There are many times that words like "approximately',''slightly", "tend", "similar" etc.. Although such words are not forbidden in the context of science, overusing them makes the data weak and the conclusions subjective.

Additional comments

Having many reservations about the content of the study due to the used English language, this study should be better presented, in the context of building a better rational and concluding to findings that are related to the introduction (give answers to open questions).

Reviewer 2 ·

Basic reporting

BASIC REPORTING
Thank you for the opportunity to review this manuscript entitled, “Strategies of elite Chinese gymnasts in coping with landing impact from backward somersault.” It was a great pleasure for me to read a paper focussing on gymnastics biomechanics and landing strategies. While I very much enjoyed the topic, I do have some concerns about some of your methods and also strongly believe that this manuscript can be significantly improved in terms of your English sentence structure and grammar. Below are my specific comments and recommendations.

Abstract:
Line 33-34: This sentence doesn’t add much to the abstract. How would this information be used for identification and selection of young gymnasts? I’m not sure how by identifying young gymnasts who use this landing technique would then transfer into these gymnasts being more successful than others that don’t use this technique. I think this research might be more useful for injury prevention, rather than talent identification.

Introduction:
In your introduction you introduce the importance of landing in gymnastics and the high loads associated with landings and that a routine always ends with a landing. While this is correct, this is mostly describing dismount landings, which are different from skill based landing as dismounts generally take place from a height (not from the ground level). I’m just not sure you have adequately justified why you investigated the standing back somersault, when you have described the potential for injury when performing dismount landings. Also, because a standing back somersault is a very basic skill, elite level gymnasts would only perform this type of skill during warm-up activities. Can you please re-work this introduction to provide more justification as to why you chose to investigate the back somersault skill?
Line 44: Add a space between body weight and (BW)
Line 49: Might be worthwhile adding in what FIG considers to be excessive knee flexion here. Then you can compare your results to what FIG considers as ideal later in your discussion. It will just add a bit more depth and context to your findings.
Line 55: I believe there has been a number of studies that have demonstrated the relationship between stiffness and injuries. Maybe another example, more closely related to gymnastics (jumping or landing) would be beneficial here.
Line 57-60: These are really good points. Please consider re-wording to just be a little clearer how there could affect landing strategies, because you really only just state them here. A bit more detail might be beneficial.
Line 61-69: Some great points raised in here, well done!
Line 68: Fix reference here, there is an extra space after the first bracket
Line 72-73: Fix reference here, Caine et al., 2003 is not in blue like the rest of the references
Line 73-74: So some included international/national level gymnasts or were they all collegiate and young gymnasts? The phrasing of this sentence is confusing, please make this line clearer.
Line 78: The use of “What else,” Is not really scientific writing, please consider changing
Line 89: Try to avoid the use of “we,” when writing scientifically
Line 92-93: This line doesn’t add anything to the introduction. If you wanted to keep this in, you would need to add more information as to how that could be used, but I would just consider removing.

Methods:
Line 96: Is a sample size of six big enough to allow for conclusive results? Did you conduct a power analysis?
Line 99: Remove capitalisation of “All” in the middle of a sentence.
Line 106-109: Is this a standard marker model? If so please refer to the model. And if not, more detail is needed about the exact location of where those markers were placed is needed.
Line 110-111: Was the mat over the force plate attached to adjoining mats, or was the mat over the force plate separate from mats in surrounding areas? If the mat over the force plate was joined to surrounding mats, then some of the force upon landing would have been distributed to the adjoining mats and would result in a reduction the GRF. Please state whether or not the mats were joined and the dimensions of the landing mat.
Line 110: Did you only analyse vGRF? Because of the rotational nature of the back somersault, it is unlikely that upon landing all the force took place in the vertical direction. Did you take into account the anterior-posterior GRF or medial-lateral GRF when creating your model, or only vGRF?
Line 122-123: So you only used one trail for each participant? Is there a reason you didn’t use all three? This could have increased the number of trials you had, which could have been beneficial considering you had a very small sample size (only six participants)
Line 125-130: How did you pick these filter cut-offs? Is there a reason why they were different?
Line 141: Avoid starting a sentence with “we”
Line 141-143: So only one trial was used for modelling and simulation? Is there a reason you didn’t use more trials? Can you be certain in your results if it is from only one trial from one participant?
Line 148: Is the mat model used similar to what was used in the actual data collection? Was the mat that they landed on over the force plate 2m x2m in size? Wouldn’t this affect results if the mat is not similar (force more dispersed on landing)?
Did any statistical analysis take place? This should be reported in the methods section.

Results:
Line 157-164: Are these results from the model simulation? Maybe just make this clearer.
Line 168-169: Please explain why a difference of 4.6% is considered good.
Discussion:
Line 182: This is out of place here, considering talking about this further into the discussion in much greater detail. Please cover how does this information enhance our understanding of gymnastics landings and how can this information be used?
Line 184-187: Great points!
Line 194-197: This line seems a bit repetitive, consider re-wording.
Line 203-205: Consider re-wording, this sentence is a bit confusing to follow.
Line 207-210 Consider merging these sentences, as they are very similar.
Line 215-217: Also they were performing very different tasks. In McNitt-Gray’s study, gymnasts were performing drop landings from different height, and in this study were performing standing back somersaults.
Line 226-228: Good point!
Line 229-231: You first state that in T2 the gymnasts actively extended their lower extremity joints, then in the next sentence say they flexed their joints during this phase. Can you please make this clearer?
Line 232-233: Could actually discuss the FIG joint flexion requirements here. What are they and did your participants meet them?
Line 234: “the EMG of lower extremity muscles still remained a higher level of activation,” compared to what?
Line 236-238: This is a really interesting point that requires more detail and discussion. This one of the main findings of your paper, so please explain this in greater detail.
I would consider adding in a paragraph about how the findings from you study can be utilised. What does this add to the current literature and how can these results be used in gymnastics?

Conclusions:
Line 261: This does not add anything to the conclusion. It is also the same sentence you have used word for word earlier in the manuscript. I would consider addressing how these findings can be used in a paragraph in the discussion, as suggested above.
Figures:
Figure 1 is a bit pixelated, is it possible to get a clearer version of this image?
Are Figure 3, 4, 5 & 6 from only one participant or mean values?

Experimental design

There a few things with require further justification or clarification in regards to your experimental design.
1. Why did you choose to investigate only the back somersault skill? I understand it is a foundation gymnastics skill, but elite gymnasts would only perform this skill in warm-up activities. Further justification is needed.
2. How did you determine the filter cut-offs you used for your data? Did you perform some kind of test to determine the optimal cut-off frequencies for your data set (i.e. residual analysis)?
3. You study had a very small sample size, which I understand can be difficult when working with elite athletes. However, you only used one trial per participant for analysis, and only one trial for the modelling. Why did you not use all available trials you collected, instead of the best trial of three?
4. Further clarification is required on the matting used during the data collection. Dimensions of the mat need to be reported and please state whether the mat placed over the FP was connected to surrounding mats as this would all impact the results.
5. Can you please provide further clarification as to why you only investigated vGRF and not GRF in all directions? Also, why did you only use vGRF for your modelling?
6. No statistical analysis was reported in the methods, please add this in.

Validity of the findings

No comment

Additional comments

No comment

Reviewer 3 ·

Basic reporting

No comment' if you have nothing to add.

Experimental design

No comment' if you have nothing to add.

Validity of the findings

No comment' if you have nothing to add.

Additional comments

In my opinion, I think in the technical execution of the backward somersault very important:
1) Correct specification of the backward somersault phasic structure:
- preparation phase - start position (launching position);
- base phase, multiplication body posture, maximum height of COM
- final phase - landing (final posture) with the two phases (initial and final).
2) Presentation of the work of arms and shoulders.
3) Angular velocity between segments, regarding the relationship between toes, knee, shoulders and arms with the COM (hip) axis of rotation.
4) Practical application and recommendation: I do not think there is a link between backward somersault landing and various acrobatic lines landing of different difficulty (tucked, picked and stretched, double back somersault – tucked, pike and tucked with 360° twist).

Annotated reviews are not available for download in order to protect the identity of reviewers who chose to remain anonymous.

---

## Round 0.2 · Minor Revisions

I would like to thank you for taking on board virtually all of the comments of the reviewers from the first mission of this manuscript to our satisfaction. Reviewer two still has a small number of very minor comments that you need to amend. Further, I also have found several instances of sentences that could be much better written. Please check these identified sections of your manuscript and look to have one final proofread of this by an expert in the English language to ensure a high standard is found within the entire manuscript.

line 55 – 57: should be rewritten to something like the following “Bradshaw and Hume (2012) investigated changes in the landing of two gymnasts over a period of eight years. Both gymnasts showed an increase in ankle plantar-flexion stiffness from 10.8 to 13.9 kN/m, with both gymnasts reporting severe pain in one or both heels over this period of time”.
Line 73: never start a sentence with “And”, so please change the sentence about the participants being collegiate students or young athletes.
Line 75 – 78: this also requires some improvements in its written English.
Line 91: this sentence should start with “It is hypothesised that elite gymnasts…”.
Line 158: this sentence should start with “This model was then validated…”.
Line 190-191: this sentence should read “and muscle activation characteristics of each phase of the backward somersault in elite gymnasts”.
Line 215 – 217: when you refer to the knee joint and hip joint on line 217, please make it clear if you’re referring to the peak angular velocities as you did with respect to the ankle joint on line 216.
Line 234 – 237: please provide a reference to who these authors were who suggest this balancing of the landing impact.
Line 251-253: this should be written as something like “Devita and Skelly(1992) suggested that during a stiff landing in which there is less than 90° of knee flexion and larger GRFs than a soft landing, the ankle plantarflexors produced a larger torque, and the ankle muscles absorbed a greater amount of the impact forces on landing”.

Reviewer 2 ·

Basic reporting

The language of this manuscript has been significantly improved throughout. It is now much clearer and easier to follow as a reader. There are only a few sentences that could benefit from a bit more editing (lines 55-58, 87-89, and 146-147). The revised introduction now discusses more relevant background literature which builds a more robust justification for your study. The addition of a hypothesis also strengthens your paper. Figures have been improved and now include much more detailed figure captions.

Experimental design

The research question has been well justified and clearly stated. The methods section has also been improved to clear up any ambiguous wording, now making it much easier for the reader to understand exactly what procedures and equipment was used. Thank you for adding in additional information about your marker set, statistical analysis and the simulation software.

Validity of the findings

Discussion of the results has been significantly improved. It is well-structured and follows a logical train of thought and links the current results to relevant literature. I particularly liked the addition of the paragraph from line 251-258 discussing how this information could be used to assist with injury prevention. Information on the practical implications of the research is very important, so it was good to see this included. The conclusions are appropriate and adequately summarise the key findings of the study.

Additional comments

Thank you for the opportunity to review this manuscript again, as this manuscript has been significantly improved since the last version. The authors appear to have taken on all comments and suggestions from the reviewers and as a result now have a much stronger and more robust manuscript. Well done!

---

## Round 0.3 · accepted · Accept

We thank the authors for attending to the required revisions and would recommend this paper now be accepted for publication.